# HORIZON-LENGTH PREDICTION:
# ADVANCING FILL-IN-THE-MIDDLE CAPABILITIES FOR CODE GENERATION WITH LOOKAHEAD PLANNING

## ABSTRACT

Fill-in-the-Middle (FIM) has become integral to code language models, enabling generation of missing code given both left and right contexts. However, the current FIM training paradigm, which reorders original training sequences and then performs regular next-token prediction (NTP), often leads to models struggling to generate content that aligns smoothly with the surrounding context. Crucially, while existing works rely on rule-based post-processing to circumvent this weakness, such methods are not practically usable in open-domain code completion tasks as they depend on restrictive, dataset-specific assumptions (*e.g., generating the same number of lines as in the ground truth*). Moreover, model performance on FIM tasks deteriorates significantly without these unrealistic assumptions.

We hypothesize that NTP alone is insufficient for models to *learn effective planning* conditioned on the distant right context, a critical factor for successful code infilling. To overcome this, we propose **Horizon-Length Prediction** (**HLP**), a novel training objective that teaches models to predict the number of remaining middle tokens (*i.e., horizon length*) at each step. HLP advances FIM with lookahead planning, enabling models to inherently learn infilling boundaries for arbitrary left and right contexts without relying on dataset-specific post-processing. Our evaluation across different models and sizes shows that HLP significantly improves FIM performance by up to 24% relatively on diverse benchmarks, across file-level and repository-level, and without resorting to unrealistic post-processing methods. Furthermore, the enhanced planning capability gained through HLP boosts model performance on code reasoning. Importantly, HLP only incurs negligible training overhead and no additional inference cost, ensuring its practicality for real-world scenarios.

## 1 INTRODUCTION

Large Language Models (LLMs) trained on massive source code data have demonstrated significant progress in coding-related tasks (Lozhkov et al., 2024; Guo et al., 2024; DeepSeek-AI et al., 2024; Hui et al., 2024). While natural language generation predominantly follows a Left-to-Right (L2R) approach, *Fill-in-the-Middle* (FIM), or *infilling*, is ubiquitous in code completion scenarios. This prevalence stems from the iterative nature of coding, which involves frequent edits and insertions rather than a single left-to-right pass (Bavarian et al., 2022; Fried et al., 2023). In an infilling task, the model is asked to generate the missing code in the middle, conditioned on both the preceding (left) and the succeeding (right) contexts.

The common practice to achieve FIM capability with uni-directional auto-regressive models is to reorder the original sequence of prefix-middle-suffix into either prefix-suffix-middle (PSM) or suffix-prefix-middle (SPM). This reordering allows the middle part to be predicted while conditioned on both left and right contexts as past tokens (Bavarian et al., 2022; Fried et al., 2023). Typically. an `<end_of_insertion>` (i.e., `<eoi>`) token is appended to the end to indicate the generation boundary. At training time, the prefix, middle, and suffix are determined by a random split, and the reordered sequence is fed to the model for standard next-token prediction (NTP).

A key challenge in FIM is to seamlessly connect the generated middle to the given suffix considering both fluency and semantics, a difficult task for models to learn in practice. As pointed out by the previous work (Bavarian et al., 2022), though model is trained to generate `<eoi>` when middle ends and connects to suffix, it often fails to do so at the right place during test time, resulting in generation that does not connect well to suffix. We hypothesize that this challenge stems from the fundamental

difference in *prediction horizon* compared to standard NTP. In NTP, the model only needs to consider a horizon of one token at a time. In contrast, FIM requires the model to plan for a much longer horizon, *i.e.,* the entire length of the missing middle section. This extended horizon is crucial because the model must generate a sequence that not only follows the left context but also smoothly transitions to the right context, which may be many tokens away. Standard NTP training does not adequately prepare models for this long-horizon planning task. Consequently, models often struggle to maintain coherence over the longer sequences required in FIM, particularly when approaching the transition to the right context. Without effective long-horizon planning, generation frequently falters towards the end of the infill, failing to create a smooth connection with the given suffix.

Most existing FIM benchmarks circumvent the above challenge to some extent by devising rule-based post-processing to truncate redundant parts at the end of the generation (Gong et al., 2024; Zhang et al., 2023; Ding et al., 2023; Wu et al., 2024), as shown in Figure 1. We argue that these post-processing techniques lack practicality as they rely on restrictive and dataset-specific assumptions, such as the exact number of lines of the expected completion (Zhang et al., 2023; Wu et al., 2024) or a specific structure to be met (Ding et al., 2023; Gong et al., 2024). Essentially, we need to develop better models that are capable of spontaneously terminating generation at the correct point with respect to arbitrary left and right contexts.

```
# SAFIM block_completion_004202
# Problem: Helping the Nature
t = int(input())
for _ in range(t):
    n = int(input())
    A = list(map(int, input().split()))
    res, r = 0, 0
    for i in range(n - 1):
        x = A[i + 1] - A[i]
        if x > 0:
            r += x
        else:
            r += abs(x)
        res += abs(x)
    res += abs(r - A[n - 1])
    print(res)
        res += abs(x)
    res += abs(r - A[n - 1])
    print(res)
```

Figure 1: An example that illustrates how post-processing truncates redundant parts at the end of generation. middle is generated by the model, which introduces syntax error and breaks correctness if directly connected to suffix. After truncating the part with strike-through through post-processing, middle successfully connects to suffix without any errors.

To this end, we propose **Horizon-Length Prediction** (**HLP**) to improve code infilling by teaching models to *plan ahead* on the number of tokens to be generated (*i.e.,* horizon length). Specifically, given the hidden state of the current token, we introduce an auxiliary training objective to predict *the number of future tokens* required to complete middle, in addition to standard next-token prediction (NTP). Unlike rule-based post-processing, HLP is generalizable as it does not require any task-specific knowledge.

Through comprehensive evaluation we demonstrate that HLP achieves up to 24% improvements relatively on diverse FIM benchmarks at both file-level and repository-level, with no access to task-specific post-processing. Moreover, with an emphasis on planning, training with HLP also helps achieve superior model performance on code reasoning. Besides, HLP is also extremely efficient as its training overhead is negligible and it doesn't have any inference cost.

Our key contributions are as follows:

- We highlight that post-processing methods adopted by current benchmarks overestimate existing code LLMs' FIM performance, and empirically quantify the gap. We further draw attention to models' *long-horizon planning* capability as the key to successful code infilling.

- We propose **Horizon-Length Prediction** (**HLP**), a novel training task that advances fill-in-the-middle capability by teaching LLMs to plan ahead over arbitrarily long horizons. HLP complements the standard next-token prediction by training LLMs to predict the remaining number of future tokens required to complete middle (i.e., horizon length).

- Our evaluation shows that HLP not only improves code infilling by up to 24% across various benchmarks without using any rule-based and/or dataset-specific post-processing, but also enhances performance on code reasoning. HLP is also super efficient as it only incurs negligible training overhead while not adding any inference overhead.

## 2 POST-PROCESSING FOR FILL-IN-THE-MIDDLE

| Post-processing Criteria | |
|---|---|
| RepoEval (Zhang et al., 2023) CrossCodeLongEval (Wu et al., 2024) | Truncate generation to **the same number of lines as in ground truth**. |
| CrossCodeEval (Ding et al., 2023) | Truncate generation **at the first complete statement**. |
| SAFIM (Gong et al., 2024) | Stop when **the target program structure in ground truth** is generated. |

Table 1: Post-processing criteria used in existing FIM benchmarks. Text in bold denotes restrictive dataset-specific knowledge they employ in evaluation.

Most existing FIM works rely on post-processing to truncate code completions generated by LLMs for infilling tasks (Gong et al., 2024; Zhang et al., 2023; Ding et al., 2023; Wu et al., 2024). While such post-processing can enhance the FIM performance, we argue that they are designed based on dataset-specific prior knowledge, and thus fundamentally limited and impractical for real-world scenarios (§2.1). Through evaluation, we show that FIM performance of existing code models drops significantly without post-processing, suggesting that post-processing conceals models' inability to determine the end of insertion (§2.2). Furthermore, we show that plausible generation of middle without careful planning can easily lead to the failure in connecting to suffix at the end, which can not be mitigated even with post-processing (§2.3). We believe that post-processing leads to an overestimation of infilling capability of existing code LLMs and more generalizable techniques are in need to advance LLMs' FIM performance.

| | SAFIM | | | | Avg |
|---|---|---|---|---|---|
| | Algo | Algo$_{v2}$ | Control | API | |
| DS-1.3B | | | | | |
| w/ post | 43.9 | 49.2 | 55.6 | 62.9 | 52.9 |
| w/o post | 39.8 | 42.4 | 52.4 | 56.1 | 47.7 |
| rel. diff | **-9.3%** | **-13.8%** | **-5.8%** | **-10.8%** | **-9.9%** |
| DS-6.7B | | | | | |
| w/ post | 54.9 | 58.9 | 68.1 | 71.0 | 63.2 |
| w/o post | 53.4 | 56.7 | 66.6 | 69.0 | 61.4 |
| rel. diff | **-2.7%** | **-3.7%** | **-2.2%** | **-2.8%** | **-2.8%** |
| SC2-3B | | | | | |
| w/ post | 48.1 | 53.5 | 60.1 | 68.4 | 57.5 |
| w/o post | 45.4 | 49.7 | 57.1 | 61.3 | 53.4 |
| rel. diff | **-5.6%** | **-7.1%** | **-5.0%** | **-10.4%** | **-7.2%** |
| SC2-7B | | | | | |
| w/ post | 50.4 | 55.8 | 62.3 | 70.3 | 59.7 |
| w/o post | 48.4 | 53.1 | 60.4 | 63.9 | 56.5 |
| rel. diff | **-4.0%** | **-4.8%** | **-3.0%** | **-9.1%** | **-5.4%** |

Table 2: Effect of post-processing techniques for different code LLMs on SAFIM, where "**w/ post**" refers to using post-processing, "**w/o post**" refers to not using post-processing, and "**rel. diff**" refers to the relative performance difference between the two. We follow the same settings used in §4.1.

### 2.1 POST-PROCESSING REQUIRES TASK-SPECIFIC KNOWLEDGE

Post-processing methods adopted by recent FIM benchmarks typically assume a certain completion type and perform rule-based truncation accordingly. Table 1 summarizes the post-processing criteria of four popular FIM benchmarks, highlighting the specific rule used for each dataset. These criteria do not transfer across datasets, nor are they generalizable to FIM in the real-world scenario where both left and right contexts can be arbitrary. Given the complexity of programming languages, it is infeasible to devise any rule-based post-processing for general code infilling in open-domain scenarios. A more practical and effective solution is to let the model itself decide when to stop by learning from the massive data.

### 2.2 POST-PROCESSING CONCEALS LLMS' INABILITY OF CONNECTING TO SUFFIX

To further demonstrate to what extent post-processing conceals LLMs' inability of connecting to suffix, we conduct a comprehensive experiment on SAFIM. We compare FIM performance of four different code LLMs, with or without post-processing. As shown in Table 2, after removing post-processing, we

see pass@1 drops by up to 13.8% across all models, revealing that post-processing seriously obfuscates LLMs' inherent inability to connect seamlessly to suffix during infilling.

## 2.3 FILL-IN-THE-MIDDLE REQUIRES PLANNING CAPABILITY OF LLMS

We argue that FIM requires planning from LLMs for deeper reasons beyond simply predicting the `<eoi>` token. Figure 2 provides an illustrative example that highlights the importance of planning ahead in FIM tasks. In this example, compared with the ground truth (*i.e.,* middle in **Reference**), the model generation (*i.e.,* middle in **Answer**) does not correctly connect to suffix. To elaborate, since the beginning of suffix is a function call (`Recognizer()`) that can only be accessed through a specific object (`speech`), the model has to end the generation with `speech.` to properly connect to suffix, but it fails to do so in **Answer**.

Interestingly, the model has demonstrated its knowledge of how to call the function by generating `speech.Recognizer()` in **Answer**. However, without careful planning, it prematurely writes this call before other necessary code (the assignment statement of `self.voice`). This example illustrates that the real challenge lies in planning the entire generation, rather than understanding individual components. Furthermore, this case demonstrates that post-processing, despite taking advantage of task-specific prior knowledge, cannot adequately address such planning failures. Truncating the code after `speech.` in **Answer** would result in loss of other necessary code for correctness. This observation underscores the irreplaceable importance of planning ahead in code infilling.

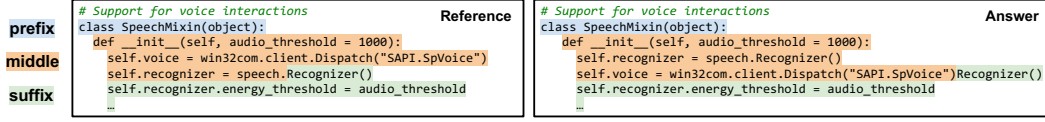

Figure 2: An example showing that FIM requires planning capabilities. middle in **Reference** refers to the ground truth and middle in **Answer** refers to the code generated by LLMs, given prefix and suffix. Compared with the ground truth, LLM fails to connect to suffix due to lack of planning capability.

## 3 HORIZON-LENGTH PREDICTION

Given a document $D = \{x_t\}_{t=1}^{T}$ that contains $T$ tokens $x_1, x_2, \cdots, x_T$, existing FIM training scheme can be divided into three steps: (1) Split the document $D$ into three parts: prefix-middle-suffix[1], (2) Construct a new FIM-style document $D'$ by reordering the three parts as prefix-suffix-middle, and (3) Conduct next-token prediction (NTP) training on the document $D'$.

Specifically, we define the three parts in document $D$ as prefix $= x_{1\cdots P}$, middle $= x_{P+1\cdots P+M}$, and suffix $= x_{P+M+1\cdots T}$. Then, the new document $D'$ will be formatted as follows:

$$D' = \text{<PRE> prefix <SUF> suffix <MID> middle <EOI>}$$

$$= \text{<PRE>}\, x_{1\cdots P}\, \text{<SUF>}\, x_{P+M+1\cdots T}\, \text{<MID>}\, x_{P+1\cdots P+M}\, \text{<EOI>} \tag{1}$$

$$\overset{\Delta}{=} y_{1\cdots T-M+3}\, x_{P+1\cdots P+M}\, \text{<EOI>},$$

where the last step re-indexes the leading tokens up until <MID> to $y_{1\cdots T-M+3}$ to focus on the FIM part, as LLMs are expected to start infilling after <MID> token and to end generation with <EOI> token to connect to suffix accurately.

Next-token prediction (NTP) training is conducted on the document $D'$, whose goal is to minimize the following cross-entropy loss (where $P_\theta$ refers to LLMs being trained):

$$L_{NTP} = -\sum_{t=1}^{T-M+2} \log P_\theta(y_{t+1}|y_{1\cdots t})$$

$$-\sum_{t=1}^{M-1} \log P_\theta(x_{P+t+1}|y_{1\cdots T-M+3}, x_{P+1\cdots P+t}) \tag{2}$$

$$-\log P_\theta(\text{<EOI>}|y_{1\cdots T-M+3}, x_{P+1\cdots P+M}).$$

---

[1]We opt to use PSM setting in this work given our base models (DeepSeek-Coder (Guo et al., 2024) and StarCoder2 (Lozhkov et al., 2024)) were both pre-trained with PSM setting. However, we expect that our method is generalizable to SPM setting as well.

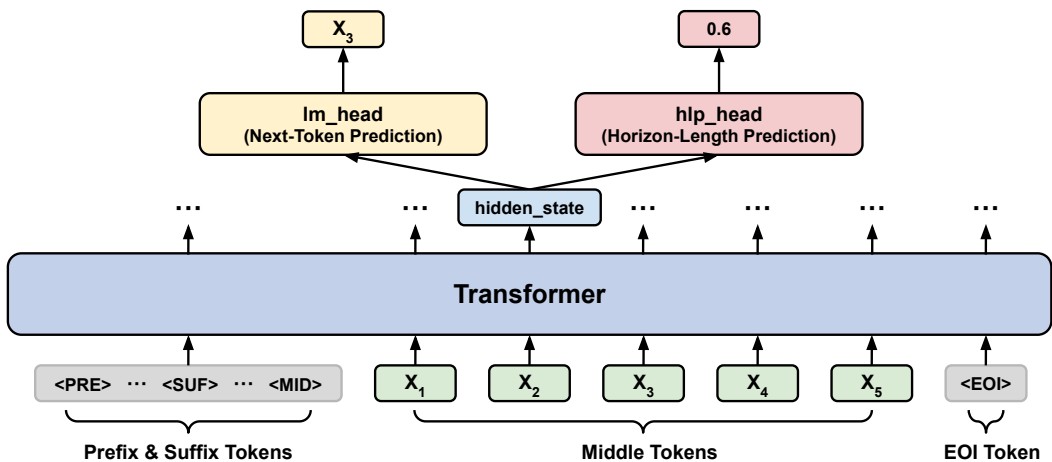

Figure 3: Overview of Horizon-Length Prediction (HLP). In this example, we set the length of middle to five tokens. Following the flow of arrows, we illustrate how the second token of middle (*i.e.,* "$x_2$") is processed through both next-token prediction objective and horizon-length prediction objective.

While NTP has provided LLMs with necessary supervision signals to learn Fill-in-the-Middle (*i.e.,* starting generation to match prefix and stopping at the right place to connect to suffix), it is shown in our analysis (§2) that LLMs trained with NTP alone do not learn this task well in practice. This is likely due to models trained with NTP only learns to predict with a horizon length of only 1 (*i.e.,* next token), and thus infilling which typically requires planning over a much longer horizon length could not be learnt effectively.

**Horizon-Length Prediction (HLP)**. To mitigate this issue, we propose adding an auxiliary training objective, namely horizon-length prediction (HLP), to improve the planning capabilities of LLMs over long horizons. Specifically, given the hidden state of current token, the model is tasked by HLP to predict the number of future tokens required to complete middle, as shown in Figure 3.

In detail, the FIM-style document $D'$ defined in Eq. (1) contains three different parts, including prefix, suffix, and middle, and HLP is applied to middle as it is the major focus of FIM tasks. Recall that middle has $M$ tokens, ranging from $x_{P+1}$ to $x_{P+M}$. At each position $t \in [1, M]$ in middle, HLP aims to predict the number of future tokens required to complete the middle, which is $M - t$. Considering that the context window size of LLMs can be infinite in theory and so is $M - t$, instead of formatting HLP as a classification task with a fixed set of discrete values as labels, we make it a regression problem by taking the normalized count of future tokens as the target:

$$y_t = \frac{M - t}{M} \in (0, 1]. \tag{3}$$

As such, the target is always within the $(0, 1]$ interval regardless of the model's context window size.

HLP is implemented as a linear layer on top of the transformer model (*i.e.,* hlp_head in Figure 3) with weight $w_{hlp}$, whose input is the hidden state $h_t$ from the last attention layer. The output $w_{hlp}^\top h_t$ is converted to a value between 0 and 1 through a sigmoid layer $\sigma$ to represent the final prediction. We use L1 loss for HLP:

$$L_{HLP} = \sum_{t=1}^{M} |\sigma(w_{hlp}^\top h_t) - y_t|. \tag{4}$$

The full training objective is a weighted sum of NTP loss and HLP loss:

$$L = L_{NTP} + \lambda \cdot L_{HLP}, \tag{5}$$

where $\lambda$ is the tunable weight. In experiments, we set $\lambda = 0.1$, which achieves good results across benchmarks empirically.

**Why should HLP work?** Theoretically, next-token prediction (NTP) is myopic in the sense that it does not take into account future constraints when predicting the immediate next token, as it only optimizes for local token-by-token decisions. While this is less of a problem for left-to-right code generation, in FIM there is always such a constraint that the middle part must connect smoothly to the right context. Standard NTP only tackles this constraint at the very last <EOI> token. However, at test time, the <EOI> token can sometimes be completely missing due to accumulated errors in preceding tokens which are

not trained with any end-of-insertion information (Gong et al., 2024). In contrast, with HLP, every token gets a sense of how far away it is from connecting to the right context (i.e., global sequence-level planning), leading to a more reliable closing of the infilling generation. In other words, while NTP only provides a sparse signal from <EOI> token prediction at the end of generation, HLP enables a consistent optimization by providing an auxiliary learning signal at every generation step.

**Overhead Analysis.** While HLP introduces the additional hlp_head during training, the number of added parameters is $< 0.01\%$ of the base model, which incurs *almost zero training overhead*. Furthermore, the additional head can be discarded during inference, leading to *no inference overhead*.

## 4 EXPERIMENTS

**Training.** We conduct continual pre-training on a set of code LLMs of different model families and sizes to validate the effectiveness of HLP. Specifically, DeepSeek-Coder-Base 1.3B/6.7B (Guo et al., 2024) and StarCoder2 3B/7B (Lozhkov et al., 2024) are involved in our experiments. We use AdamW (Loshchilov & Hutter, 2019) as the optimizer with $\beta_1 = 0.9$ and $\beta_2 = 0.95$. We use a cosine learning rate scheduler with a peak learning rate equal to that at pre-training end. All models are trained for 200K steps with a warm-up period over the first 3,000 steps. The global batch size is 512.

**Dataset.** We use a subset of `the-stack-v2-train-smol` (Lozhkov et al., 2024) for continual pre-training which includes Python, Java, C++, and C#. In line with existing works (Guo et al., 2024; Lozhkov et al., 2024), FIM rate is set to $0.5$. We employ Best-fit Packing (Ding et al., 2024) to group multiple files into each training sequence while masking out cross-file attention. The prefix-middle-suffix split is applied to each file independently rather than the whole training sequence.

We conduct controlled experiments for all the studied code LLMs in our experiments. Specifically, we conduct two continual pre-training experiments for each model as follows:

- **NTP**: existing pre-training scheme with next-token prediction (NTP) objective only.
- **NTP + HLP**: our newly proposed pre-training scheme that incorporates horizon-length prediction (HLP) objective with next-token prediction (NTP) objective.

Throughout this section, we determine the end of generation solely based on <eoi> predicted by the model during evaluation, without any rule-based post-processing, unless otherwise specified in §4.4. We also conduct statistical analysis to validate our findings. Specifically, we conduct paired t-tests between the baseline (NTP) and our approach (NTP +HLP) across all benchmarks. We denote results that are statistically significant ($p < 0.05$) by a superscript *.

### 4.1 SYNTAX-AWARE AND MULTILINGUAL CODE FILL-IN-THE-MIDDLE

We use SAFIM (Gong et al., 2024), a syntax-aware and multilingual code Fill-in-the-Middle benchmark, to evaluate the effectiveness of HLP. SAFIM focuses on syntax-aware completions of program

| | SAFIM | | | | Average |
|---|---|---|---|---|---|
| | Algorithmic | Algorithmic$_{v2}$ | Control | API | |
| DS-1.3B | 39.8 | 42.4 | 52.4 | 56.1 | 47.7 |
| + HLP | **41.3** | **46.1** | **53.4** | **59.0** | **50.0**[*] |
| DS-6.7B | 53.4 | 56.7 | 66.6 | 69.0 | 61.4 |
| + HLP | **53.5** | **57.4** | **66.9** | **69.7** | **61.9**[*] |
| SC2-3B | 45.4 | 49.7 | 57.1 | 61.3 | 53.4 |
| + HLP | **47.2** | **52.1** | **58.7** | **64.5** | **55.6**[*] |
| SC2-7B | 48.4 | 53.1 | 60.4 | 63.9 | 56.5 |
| + HLP | **49.4** | **54.5** | **61.8** | **65.8** | **57.9**[*] |

Table 3: Pass@1 results of training w/o and w/ HLP for different code LLMs on SAFIM (Gong et al., 2024) computed with greedy decoding. We denote results that are statistically significant ($p < 0.05$) in the "Average" column by a superscript *. The same notation applies hereafter.

structures, covering algorithmic block (*i.e.,* Algo and Algo$_{v2}$), control-flow expression (*i.e.,* Control), and API function call (*i.e.,* API). It consists of 17,720 examples from four different programming languages, including Python, Java, C++, and C#. SAFIM employs execution-based evaluation and reports pass@1 as the evaluation metric. As shown in Table 3, compared with NTP only, adding HLP achieves up to 5% improvements on average across all the studied code LLMs. Specifically, HLP consistently improves LLMs' performance on completions of various program structures. Furthermore, since SAFIM is a multilingual benchmark, our evaluation also shows that incorporating HLP can improve FIM performance across different languages.

## 4.2 REPOSITORY-LEVEL CROSS-FILE CODE FILL-IN-THE-MIDDLE

| | CrossCodeEval / CrossCodeLongEval | | | | | | Average | |
| | Line | | Chunk | | Function | | | |
| | EM | ES | EM | ES | EM | ES | EM | ES |
|---|---|---|---|---|---|---|---|---|
| DS-1.3B | 15.23 | 49.64 | 22.48 | 56.40 | 4.58 | 33.96 | 14.10 | 46.67 |
| + HLP | **18.99** | **55.47** | **24.32** | **58.77** | **5.12** | **35.25** | **16.14*** | **49.83*** |
| DS-6.7B | 26.23 | 62.07 | 28.90 | 62.37 | **7.50** | **41.42** | 20.88 | 55.29 |
| + HLP | **27.35** | **63.54** | **30.08** | **63.18** | 7.22 | 40.99 | **21.55*** | **55.90*** |
| SC2-3B | 24.17 | 59.89 | 22.20 | 52.69 | 6.80 | 38.13 | 17.72 | 50.24 |
| + HLP | **25.67** | **62.62** | **30.66** | **62.01** | **7.18** | **39.42** | **21.17*** | **54.68*** |
| SC2-7B | 26.00 | 61.68 | 27.14 | 56.52 | 7.66 | 39.54 | 20.27 | 52.58 |
| + HLP | **27.58** | **63.84** | **32.86** | **64.07** | **8.44** | **41.03** | **22.96*** | **56.31*** |

| | RepoEval | | | | | | Average | |
| | Line | | API | | Function | | | |
| | EM | ES | EM | ES | EM | ES | EM | ES |
|---|---|---|---|---|---|---|---|---|
| DS-1.3B | 24.50 | 50.42 | 18.81 | 58.15 | 3.96 | 29.73 | 15.76 | 46.10 |
| + HLP | **27.25** | **53.45** | **21.81** | **59.79** | **5.93** | **31.92** | **18.33*** | **48.39*** |
| DS-6.7B | 26.62 | 52.59 | 22.69 | 61.94 | 7.47 | 36.24 | 18.93 | 50.26 |
| + HLP | **30.31** | **55.97** | **25.12** | **63.06** | **7.69** | **37.22** | **21.04*** | **52.08*** |
| SC2-3B | 21.88 | 46.74 | 18.81 | 56.66 | 4.40 | 29.99 | 15.03 | 44.46 |
| + HLP | **26.56** | **50.56** | **23.06** | **61.02** | **7.25** | **33.79** | **18.96*** | **48.46*** |
| SC2-7B | 27.94 | 51.60 | 21.56 | 58.98 | 6.81 | 32.80 | 18.77 | 47.79 |
| + HLP | **34.19** | **57.29** | **27.31** | **63.04** | **8.35** | **35.40** | **23.28*** | **51.91*** |

Table 4: Exact Match (EM) and Edit Similarity (ES) results of training w/o and w/ HLP for different code LLMs on CrossCodeEval (Ding et al., 2023), CrossCodeLongEval (Wu et al., 2024), and RepoEval (Zhang et al., 2023) using greedy decoding, following the experimental setting of existing work (Wu et al., 2024). Our evaluation is conducted under "Retrieval" mode, where evaluation prompts are constructed by prepending the retrieved cross-file context to the current file, to show the performance of repository-level cross-file code completion.

In addition to single-file FIM evaluation with SAFIM, we also evaluate the effectiveness of HLP on repository-level code Fill-in-the-Middle in cross-file scenarios via CrossCodeEval (Ding et al., 2023), CrossCodeLongEval (Wu et al., 2024), and RepoEval (Zhang et al., 2023). CrossCodeEval (Python) and CrossCodeLongEval are two repository-level cross-file benchmarks that leverage more than 1500 raw Python repositories to construct 12,665 examples across line, chunk, and function completion tasks, which are used for a more rigorous evaluation. RepoEval is another repository-level cross-file code completion benchmark consisting of 3,655 line, API, and function completion tasks created from 32 Python repositories. We follow existing work (Wu et al., 2024) to evaluate the model's FIM performance on these benchmarks and use Exact Match (EM) and Edit Similarity (ES) as our evaluation

metrics. As shown in Table 4, adding HLP provides consistent improvements for all models across different benchmarks and completion tasks. Specifically, HLP achieves up to 24% improvements on EM and 9% improvements on ES relatively, showing its significant effectiveness.

### 4.3 CODE FIXING VIA FILL-IN-THE-MIDDLE

We use Defects4J (Just et al., 2014) to evaluate the performance of HLP on code fixing. Defects4J consists of open-source bugs found across 15 Java repositories. Following existing works (Xia et al., 2023; Xia & Zhang, 2023), we collect 313 single-hunk bugs from Defects4J that can be fixed by replacing or adding a continuous code hunk. Specifically, for each bug, models are prompted to generate the correct code hunk (*i.e.,* patch) given the left and right contexts of the buggy code hunk, and the whole test suite of the project will be executed to evaluate the correctness of the generated patch. Patches that can successfully pass the test suite are referred to as *plausible patches* and we report the number of plausible patches as our evaluation metric. As shown in the "Code Fix" section of Table 5, adding HLP during training results in relatively up to 18% more bugs fixed by the model[2].

### 4.4 CODE REASONING VIA FILL-IN-THE-MIDDLE

Lastly, we examine whether HLP improves model performance on code reasoning beyond ordinary completion use cases. The motivation behind is that HLP aims to teach the model to plan ahead, and planning is a special subset of reasoning that requires an action sequence over a long time horizon (Kang et al., 2024). Towards this end, we consider CRUXEval (Gu et al., 2024) which comprises 800 Python functions paired with two distinct tasks: CRUXEval-I, where LLMs need to predict the input from the known output, and CRUXEval-O, where LLMs are required to predict the output based on the given input.

We reformat prompts of CRUXEval-I into FIM style and leave CRUXEval-O as L2R generation, both of which are evaluated in zero-shot setting. Different from previous subsections

|  | Code Fix | Code Reasoning | |
|---|---|---|---|
|  | Defects4J | CRUXEval-I | CRUXEval-O |
| DS-1.3B | 33 | 42.0 | 31.0 |
| + HLP | **39** | **44.7**[*] | **31.8**[*] |
| DS-6.7B | 58 | 52.1 | 39.2 |
| + HLP | **59** | **52.4** | **39.6** |
| SC2-3B | 39 | 42.8 | 32.1 |
| + HLP | **41** | **43.9**[*] | **32.6**[*] |
| SC2-7B | 41 | 44.4 | 35.9 |
| + HLP | **47** | **45.5**[*] | **36.1**[*] |

Table 5: Code fixing and reasoning performance of models trained w/o and w/ HLP for different code LLMs on Defects4J and CRUXEval. On Defects4J. We report the number of plausible patches under greedy decoding. On CRUXEval, we follow the original setting to do sampling with $T = 0.2$ and $n = 10$ and to extract accurate input/output values from raw generation.

where post-processing is not used, we follow the same pipeline as in the original CRUXEval paper to extract accurate input/output values from generation because we are focusing on evaluating the reasoning capability of LLMs rather than their capability of generating correct code[3]. As shown in the "Code Reasoning" section of Table 5, HLP demonstrates up to 6% improvements on both CRUXEval-I and CRUXEval-O tasks for all the code LLMs consistently, which shows that HLP also improves code reasoning capabilities of LLMs.

## 5 DISCUSSION

### 5.1 COMPARING HLP WITH MULTI-TOKEN PREDICTION

As discussed in §6, we argue that multi-token prediction (Gloeckle et al., 2024) is insufficient for planning in FIM, because multi-token prediction only enhances models' planning capability for a short

---

[2]Note that Defects4J is a small dataset with only 313 examples and models can only solve 30-60 out of those, which makes it hard to obtain statistically significant differences with greedy decoding.

[3]In CRUXEval-I, we only want to evaluate the correctness of the input value infilled by LLMs in the given assertion. However, FIM-style prompts we use in the experiments does not restrict LLMs from writing multiple assertions before starting infilling the given assertion, which is useless in this task. So we use post-processing techniques to extract the input value infilled for the given assertion to better evaluate the reasoning capabilities.

| | SAFIM | | | | Average |
|---|---|---|---|---|---|
| | Algorithmic | Algorithmic$_{v2}$ | Control | API | |
| DS-1.3B | 39.8 | 42.4 | 52.4 | 56.1 | 47.7 |
| + HLP | **41.3** | **46.1** | **53.4** | **59.0** | **50.0** |
| + multi-token prediction | 38.1 | 41.3 | 51.7 | 55.8 | 46.7 |

Table 6: Pass@1 results of training w/ HLP and w/ multi-token prediction for DeepSeek-Coder-Base 1.3B on SAFIM (Gong et al., 2024) computed with greedy decoding.

and limited horizon, which does not suites FIM well as the connection from middle to suffix happens over a long horizon. Instead, HLP focuses on long-horizon planning and is more effective for FIM. We conduct an experiment following the same settings in §4 to compare HLP with multi-token prediction on DeepSeek-Coder-Base 1.3B by predicting the next 4 tokens (Gloeckle et al., 2024). We report their performance on SAFIM. As shown in Table 6, while adding HLP to NTP largely improves the model's performance on SAFIM, multi-token prediction fails to do so. These results provide empirical evidence that long-horizon planning capabilities brought by HLP is essential for advancing FIM performance.

## 5.2 NEXT-TOKEN PREDICTION ALONE CANNOT YIELD HORIZON AWARENESS

We argue that NTP alone does not grant code LLMs the awareness of prediction horizon. Specifically, we show that hidden states of baseline models trained with NTP only do not carry information about the number of future tokens (*i.e.,* horizon length). Consequently, including HLP in training is essential for models to be knowledgeable of prediction horizon.

We design a probing task by fitting linear regression models over hidden states of code LLMs trained with or without HLP respectively, while freezing all parameters of the underlying transformers. In our experiments, we use different models to generate hidden states for 20K code snippets, which gives hidden state vectors for 7.8M tokens from the middle part. We split them into the training and the test set, ensuring no overlap at sequence level. Taking these hidden state vectors as inputs and the true normalized remaining token counts as targets, we fit two linear regression models for each code LLM trained with or without HLP, respectively.

We plot the predicted percentage of remaining tokens versus the normalized token position in Figure 4, and report the coefficient of determination ($R^2$) in Table 7. As shown, the regression model

| | Training ↑ | Test ↑ |
|---|---|---|
| DS-1.3B | 0.455 | 0.440 |
| + HLP | **0.919** | **0.915** |
| DS-6.7B | 0.525 | 0.519 |
| + HLP | **0.919** | **0.913** |
| SC2-3B | 0.364 | 0.356 |
| + HLP | **0.927** | **0.932** |
| SC2-7B | 0.418 | 0.410 |
| + HLP | **0.929** | **0.932** |

Table 7: Probing results of models trained w/o and w/ HLP. We report the coefficient of determination ($R^2$) of prediction, which is the higher the better.

does not fit well with hidden states from the baseline model, indicating that those hidden states are not strongly correlated with future token count and do not contain information about horizon length. In contrast, with HLP, the hidden states perform much better. The result demonstrates that horizon awareness does not naturally exist in language models trained with NTP, and can only be obtained through targeted training tasks such as HLP.

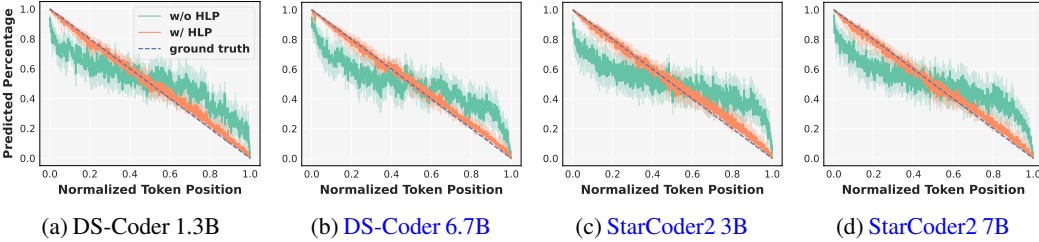

| (a) DS-Coder 1.3B | (b) DS-Coder 6.7B | (c) StarCoder2 3B | (d) StarCoder2 7B |

Figure 4: Predicted percentage of remaining future tokens (as defined in Eq. (3)) from models trained w/o and w/ HLP at different token positions, where the concrete position of each token is normalized to the corresponding percentage over the whole sequence.

## 6 RELATED WORK

**Fill-in-the-Middle for Code Language Models** The unprecedented success of causal language models such as GPT-3 (Brown et al., 2020) in natural language has inspired researchers to develop similar decoder-only models for programming languages. These models are trained on massive source code data for applications such as code generation. While early models such as Codex (Chen et al., 2021) and CodeGen (Nijkamp et al., 2023) only support Left-to-Right (L2R) generation, Fill-in-the-Middle (or infilling) has attracted increased attention because right context naturally carries an indispensable part of information for completing code in the middle (Fried et al., 2023; Bavarian et al., 2022). Subsequently, FIM training has become a common practice widely adopted by most code LLMs, such as StarCoder (Li et al., 2023; Lozhkov et al., 2024), DeepSeek-Coder (Guo et al., 2024; DeepSeek-AI et al., 2024), and Code Llama (Rozière et al., 2023).

Existing models generally tackle the infilling problem by breaking a code snippet into prefix-middle-suffix, and reordering them into prefix-suffix-middle (PSM) or suffix-prefix-middle (SPM). The transformed sequences are fed to the model during training for standard next-token prediction (NTP). We point out that the infilling task cannot be effectively learned with NTP alone, as it requires planning capability for the model to fluently and meaningfully connect the middle completion to the suffix through forward looking during auto-regressive decoding.

An alternative approach is to train two language models in different directions, with one generating from left to right and the other from right to left, and have the two generations meet in the middle (Nguyen et al., 2023). Nevertheless, the L2R model does not have access to the right context, and vice versa, which impedes holistic planning that takes into account the context from both sides.

**Planning and Lookahead in Language Generation** Standard decoder-only models are trained with next-token prediction and used to sequentially predict one token at a time, conditioned only on past tokens, in an auto-regressive manner. One drawback of this paradigm is that models are not aware of future tokens during decoding. The token that maximizes the conditional probability at current step may lead to suboptimal continuation, and consequently the model can fail to compose a fluent and sensible generation that meets human requirements. Various decoding techniques have been proposed to address the problem through tree search with lookahead heuristics, particularly for constrained generation problems (Lu et al., 2022; Huang et al., 2024). While these methods are training-free, they inevitably incur additional cost of inference complexity.

Apart from those, Gloeckle et al. (2024) proposed to predict multiple tokens from a single hidden state during both training and inference, which was shown to achieve stronger performance on coding tasks with no computation overhead. While multi-token prediction enhances models' planning capability within the $n$ tokens predicted together ($n \le 8$), we argue that with a small $n$, the limited horizon is usually insufficient for planning in the case of infilling as the connection from middle to suffix only happens towards the end of the generation. In contrast, HLP adopts a global and arbitrary long horizon over all future tokens by counting the remaining generation budget, which more effectively helps models to close the generation fluently with early planning.

## 7 CONCLUSION

Fill-in-the-Middle is ubiquitous in code completion, and therefore, has become an important consideration in the development of code language models. The current FIM training paradigm splits and reorders original training sequences (prefix-middle-suffix) into FIM-style sequences (prefix-suffix-middle/PSM or suffix-prefix-middle/SPM), and performs standard next-token prediction. However, this approach frequently results in models struggling to generate content that smoothly aligns with the right context. While existing FIM benchmarks frequently rely on different post-processing methods to circumvent this problem, we emphasize that such methods typically require dataset-specific assumptions, which are impractical in real-world scenarios. To address this limitation and enhance the infilling capability of code language models, we propose **Horizon-Length Prediction** (**HLP**). HLP teaches models to predict the portion of remaining tokens at every step. Experiments across different model families and sizes show that HLP improves infilling performance on diverse FIM benchmarks, across file-level and repository-level, and without using any dataset-specific post-processing. Moreover, the enhanced planning capability acquired through HLP training also boosts models' performance on code reasoning tasks, suggesting that HLP may broadly improve language models' reasoning capabilities. Besides, HLP is also efficient as it does not cause any inference overhead and the training overhead is negligible as well. Our work marks a significant advancement in developing more effective code language models for real-world applications.

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

## A APPENDIX

### A.1 EFFECT OF HLP ON LEFT-TO-RIGHT PERFORMANCE

While HLP have significantly improved the FIM performance of LLMs, we also study its impact on the L2R code completion. To this end, we evaluate L2R performance on HumanEval (Chen et al., 2021) and MBPP (Austin et al., 2021) with DeepSeek-Coder-Base 1.3B. We further employ HumanEval+ and MBPP+ from EvalPlus Liu et al. (2023) for more rigorous evaluation with better test coverage. As shown in Table 8, with HLP applied to FIM data only (*i.e.,* HLP$_{\text{FIM}}$), the performance on L2R tasks sometimes shows a slight degradation. We hypothesize that applying HLP to middle only causes unbalanced training on prefix and suffix parts.

|  | Left-to-Right | | Fill-in-the-Middle |
| --- | --- | --- | --- |
|  | HumanEval (+) | MBPP (+) | SAFIM |
| DS-1.3B | **26.3 (22.0)** | **45.8 (36.7)** | 47.7 |
| + HLP $_{\text{FIM}}$ | 25.5 (21.3) | 45.8 (36.5) | **50.0** |
| + HLP $_{\text{FIM}}$ + HLP $_{\text{L2R}}$ | **26.2 (22.0)** | 45.7 (36.6) | 49.6 |

Table 8: Effect of HLP$_{\text{FIM}}$ only and HLP$_{\text{FIM}}$+HLP$_{\text{L2R}}$ for DeepSeek-Coder-Base 1.3B on L2R and FIM tasks. On L2R tasks including HumanEval (+) and MBPP (+), we do sampling with $T = 0.8$ and $n = 200$. We report pass@1 performance of all the models, where numbers outside and inside parenthesis "()" indicate `base` and `plus` versions of EvalPlus, respectively. For FIM experiments on SAFIM, we follow the same settings used in §4.1.

To mitigate such effect, we need to devise another HLP task that can be applied to L2R training (*i.e.,* HLP$_{\text{L2R}}$). However, the original design of HLP in §3 is not directly applicable to L2R data. While the end of middle in FIM data is strictly bounded by the beginning of suffix, the end of L2R data does not have any clear signals, as it is often possible to add additional contents (*e.g.,* another line of code or a new helper function) to the end of document fluently without any restrictions.

Therefore, instead of taking the entire code file as prediction horizon, we ask the model to predict the number of future tokens **required to complete current line** in L2R training, which is a natural semantic unit in code. Furthermore, to avoid conflicts between HLP$_{\text{FIM}}$ and HLP$_{\text{L2R}}$, we use two independent hlp_heads to let the model learn HLP$_{\text{FIM}}$ and HLP$_{\text{L2R}}$ separately. As shown in Table 8, by applying HLP$_{\text{FIM}}$ and HLP$_{\text{L2R}}$ simultaneously, the performance degradation on L2R tasks is recovered, with the improvement on FIM tasks largely retained. These results demonstrate the generalizable effectiveness of HLP and shows the huge potential of applying the idea of HLP to more general training scenarios.

## A.2  ABLATION STUDIES

We conduct several ablation studies to justify the design choices of HLP. In this section, we conduct experiments using DeepSeek-Coder-Base 1.3B, follow the same settings in §4, and report models' performance on SAFIM.

**Complexity of** hlp_head**.** We conduct an experiment to study the effect of the complexity of hlp_head by replacing the original linear layer (*i.e.,* "HLP (linear)") with a two-layer MLP with ReLU (*i.e.,* "HLP (mlp)"). As shown in Table 9, increasing the complexity of hlp_head does not bring significant improvements. We have also conducted a paired t-test between "HLP (linear)" and "HLP (mlp)" and did not see any clear directional statistical significance between them. Such results indicate that the complexity of hlp_head does not have a major impact on performance.

|  | SAFIM | | | | Average |
| --- | --- | --- | --- | --- | --- |
|  | Algorithmic | Algorithmic$_{\text{v2}}$ | Control | API | |
| DS-1.3B | 39.8 | 42.4 | 52.4 | 56.1 | 47.7 |
| + HLP (linear) | 41.3 | **46.1** | 53.4 | **59.0** | **50.0** |
| + HLP (mlp) | **41.6** | 45.9 | **54.0** | 57.4 | 49.7 |

Table 9: Pass@1 results of HLP w/ linear layer as hlp_head (*i.e.,* "HLP (linear)") and w/ MLP layer as hlp_head (*i.e.,* "HLP (mlp)") for DeepSeek-Coder-Base 1.3B on SAFIM (Gong et al., 2024) computed with greedy decoding.

**Applying HLP to all tokens *v.s.* first token only.** While it is easy to see that knowing the HLP loss on the first token is sufficient to infer the horizon length in theory, having HLP loss on every token provides denser and consistent supervision signals which makes learning easier (as discussed in §3). It also helps regularize the hidden representation of every subsequent token. To empirically show this, we conduct an experiment by only applying HLP to the first token only (i.e., "HLP (first)") and compared its performance with our original HLP design (i.e., "HLP (all)"). As shown in Table 10, while applying HLP only to the first token performs better than NTP only, applying HLP loss for each token can achieve better performance than applying it to just the first token.

| | SAFIM | | | | Average |
|---|---|---|---|---|---|
| | Algorithmic | Algorithmic$_{v2}$ | Control | API | |
| DS-1.3B | 39.8 | 42.4 | 52.4 | 56.1 | 47.7 |
| + HLP (all) | **41.3** | **46.1** | **53.4** | **59.0** | **50.0** |
| + HLP (first) | 40.0 | 44.4 | 52.3 | 57.4 | 48.5 |

Table 10: Pass@1 results of applying HLP to all tokens (*i.e.,* "HLP (all)") and first token only (*i.e.,* "HLP (first)") for DeepSeek-Coder-Base 1.3B on SAFIM (Gong et al., 2024) computed with greedy decoding.

**Normalized *v.s.* unnormalized targets.** We use normalized targets in our original HLP design (*i.e.,* using $\frac{M-t}{M}$ rather than $M - t$) is that the scale of HLP loss will be otherwise in a huge range, e.g., some examples has single digit loss while some might have thousands. To further study the effect of normalization, we conduct an experiment by using $M - t$ as the target (*i.e.,* "HLP (raw)") rather than $\frac{M-t}{M}$ (*i.e.,* "HLP (normalized)"). To achieve this, we remove the sigmoid function from the original HLP. As shown in Table 11, setting the target as $M - t$ fails to improve FIM performance, likely due to the large-scale HLP loss after using $M - t$ as the target interferes with NTP pre-training.

| | SAFIM | | | | Average |
|---|---|---|---|---|---|
| | Algorithmic | Algorithmic$_{v2}$ | Control | API | |
| DS-1.3B | 39.8 | 42.4 | 52.4 | 56.1 | 47.7 |
| + HLP (normalized) | **41.3** | **46.1** | **53.4** | **59.0** | **50.0** |
| + HLP (raw) | 27.7 | 29.8 | 34.4 | 47.7 | 34.9 |

Table 11: Pass@1 results of using normalized targets (*i.e.,* "HLP (normalized)") and unnormalized targets (*i.e.,* "HLP (raw)") in HLP for DeepSeek-Coder-Base 1.3B on SAFIM (Gong et al., 2024) computed with greedy decoding.

