# OpenReview forum: "Horizon-Length Prediction: Advancing Fill-in-the-Middle Capabilities for Code Generation with Lookahead Planning"
_ICLR.cc/2025/Conference — Submitted to ICLR 2025_

### Official Review · Reviewer_ogj5 · 2024-10-28

**Soundness:** 3
**Presentation:** 3
**Contribution:** 3
**Rating:** 5
**Confidence:** 3

**Summary:**

This paper discusses the problem of fill-in-the-middle capabilities of code generation. The next-token-prediction way tends to make the model fail in long-length code generation to combine with both the left and right code sides. The paper proposes the method Horizon-Length Prediction (HLP) to tackle this issue. During the training, the model is also required to predict the ratio of the left token length to be generated to the total token length to be generated. This method is supposed to enhance the ability of LLMs to take care of the code generation length.

The experiments on extensive settings prove the effectiveness. The authors also use probing method to show that HLP trained model tends to have better capability on code generation length prediction. I am not an expert in code LLMs. Hence, I am not sure whether the proposed question and method are novel. Based on the current experiments, I think the results are effective and the paper is clear. However, the paper lacks extensive analysis on the proposed method like why should set like M-T/M for length prediction not other settings. I am giving a borderline in this case and will surely look through other reviews and authors' comments for final evaluation.

**Strengths:**

The issue and the method are clearly explained, the experiments are carried out extensively and achieved notably better performance. Meanwhile, the method will nearly not increase the burden of LLM training. The method is intuitively reasonable.

**Weaknesses:**

1) The techniques used are relatively simple. Most contents in the paper are well-known. However, it makes sense.

2) The setting of target as M-t/M is not such solid. In this equation, the target y also depends on the total length M. Whether it is optimal setting should be discussed. For example, ablation studies on other settings.

3) The paper also lacks the thorough analysis like why enhancing the ability to predict code generation length is such effective, especially in fill-in-the-middle problem. Will it also work in uni-directional code generation?

typo: Line 132 are are -> are

**Questions:**

As discussed in the above.

---

> ### Author Response · Authors · 2024-11-21
>
> Thank you for the valuable comments and suggestions! We put our responses as follows.
>
> > W1: The techniques used are relatively simple. Most contents in the paper are well-known. However, it makes sense. / Summary: ... not sure whether the proposed question and method are novel ...
>
> We appreciate your comment. While the underlying techniques may appear straightforward, we view this simplicity as a fundamental strength of our approach. This is particularly the case for LLM pre-training considering its expensive cost, as simplicity very often yields scalability, compatibility with various training frameworks, and robustness to hyperparameter tuning.
>
> We also respectfully disagree that *most contents in the paper are well-known*. To the best of our knowledge, the idea of HLP is novel and unexplored in LLM literature.
>
> > W2: The setting of target as M-t/M is not such solid. In this equation, the target y also depends on the total length M.
>
> Thank you for the note.  We interpret the question as whether the target should be normalized by M or not, and also kindly ask for further clarification from the reviewer in case we misunderstood. The reason why we want to do normalization is that the scale of HLP loss will be otherwise in a huge range, e.g., some examples has single digit loss while some might have thousands. To further study the effect of normalization, we have conducted an additional ablation study on DeepSeekCoder-1.3B by using M-t as the target (i.e., "HLP (raw)"), which removes its dependence on M. To achieve this, we remove the sigmoid function from the original implementation of HLP. As shown in the table, compared with the original target (i.e., "HLP (normalized)"), setting the target as M-t fails to improve FIM performance, likely due to the large-scale HLP loss after using M-t as the target interferes with NTP pre-training.
>
> | | Algorithmic | Algorithmic_v2 | Control | API | Average |
> | -------- | -------- | -------- | -------- | -------- | -------- |
> | DS-1.3B | 39.8 | 42.4 | 52.4 | 56.1 | 47.7 |
> |  + HLP (normalized) | **41.3**| **46.1** | **53.4** | **59.0**  | **50.0** |
> |  + HLP (raw) | 27.7 | 29.8 | 34.4 | 47.7 | 34.9 |
>
> > W3 (part-1): The paper also lacks the thorough analysis like why enhancing the ability to predict code generation length is such effective, especially in fill-in-the-middle problem.
>
> We appreciate this important question. The effectiveness of HLP can be explained as follows:
>
> Intuitively, next-token prediction (NTP) is myopic in the sense that it does not take into account future constraints when predicting the immediate next token, as it only optimizes for local token-by-token decisions. While this is less of a problem for left-to-right code generation, in FIM there is always such a constraint that the middle part must connect smoothly to the right context. Standard NTP only tackles this constraint at the very last <eoi> token. However, at test time, the <eoi> token can sometimes be completely missing due to accumulated errors in preceding tokens which are not trained with any end-of-insertion information. In contrast, with HLP, every token gets a sense of how far away it is from connecting to the right context (i.e., global sequence-level planning), leading to a more reliable closing of the infilling generation. In other words, while NTP only provides a sparse signal from <eoi> token prediction at the end of generation, HLP enables a consistent optimization by providing an auxiliary learning signal at every generation step.
>
> Empirically, this theoretical advantage translates to practical benefits. We empirically verified our method across a wide range of coding tasks and demonstrated its effectiveness.
>
> We hope this discussion is clearer and we will expand this discussion in the paper to make these connections explicit. Thanks again for the suggestion.
>
> > W3 (part-2): Will it also work in uni-directional code generation?
>
> Applying the same idea of HLP to uni-directional code generation is definitely interesting to explore. However, the “horizon length” in uni-directional code generation is not well-defined. As discussed in the appendix, the major difference between FIM and uni-directional code generation is that, while the length of the middle in FIM is strictly bounded by the beginning of the suffix, there is no such clear boundary in uni-directional code generation. One may think of “end of file (EOF)”, but we argue that EOF as a boundary can be ambiguous as it is often possible to add additional contents (e.g., another class or a new helper function) to the end of document fluently without any restrictions. Therefore, we exercised our first attempt with line-level boundaries for uni-directional code generation (as shown in the appendix) and left further explorations as future work.

---

> ### Author Response · Authors · 2024-11-25
> **Looking forward to your feedback**
>
> Dear Reviewer ogj5,
>
> We greatly appreciate your time and expertise in reviewing our work. We have provided additional experiments, theoretical explanations, and detailed discussions in our response based on your valuable comments.
>
> As the discussion period draws close, we greatly value your further input on our responses. We are committed to ensuring that our rebuttal aligns with your suggestions and incorporates your feedback to enhance the quality of our work.
>
> Once again, we extend our deepest gratitude for your insightful and valuable comments.
>
> Thank you and best regards.

---

> ### Comment · Reviewer_ogj5 · 2024-11-26
> **Thanks for the detailed responses**
>
> Most of my questions have been addressed. However, I tend to agree with one point raised by reviewer s26Y that requiring length prediction will limit the generalizability of the training process. For instance, current Code LLMs also mostly focus on uni-directional prediction, e.g., only function head or task description added. In this case, if this method can not be integrated with uni-direction code generation, the whole Code LLM can only be trained for FIM tasks, limiting generalization.
>
> Following the above question, between FIM and uni-directional generation, there should be some middle states. For example, the tail contents with enough information (output and several lines of codes), with few information (just output or just the final line), with no contents (uni-directional). Then, the proposed method should also perform differently across these situations. Comparing these states should enhance the understanding behind this method.
>
> I tend to keep my overall evaluation. Thank you.

---

> > ### Author Response · Authors · 2024-12-01
> > **Looking forward to further discussion!**
> >
> > Dear Reviewer ogj5,
> >
> > Thank you for your response! As the discussion period deadline approaches, we would greatly appreciate your feedback on our further clarifications for your remaining concerns.
> >
> > Your feedback would greatly contribute to our work and the ICLR community!
> >
> > Thank you and best regards.

---

> ### Author Response · Authors · 2024-11-27
> **Thank you for the response and let us further clarify the details**
>
> Thank you for your response! We would like to make further clarifications to address your concern on generalizability of our proposed method. In particular, we want to highlight that HLP is designed to be broadly *applicable for pre-training general-purpose code LLMs that can do both left-to-right generation and fill-in-the-middle*. In Appendix A.1, we discussed how to achieve *monotonic improvement* with a modified version of HLP, so that the resulting model shows superior performance on FIM tasks while retaining full capability on left-to-right (a.k.a. uni-directional) generation. Concretely, we apply HLP not only for FIM, but also for L2R training by asking the model to *predict the number of future tokens required to complete the current line*. The detailed evaluation results can be found in Table 8.
>
> Given the prevalence of FIM in code LMs and real-world code completion scenarios (which we discussed with references in our original response to W1), we believe the monotonic improvement on FIM without regression on left-to-right capability represents a major contribution of our work.
>
> Once again we greatly appreciate your feedback. Kindly please let us know if there’s anything that we can clarify further, and we are more than happy to continue the discussion.

---

### Official Review · Reviewer_s26Y · 2024-11-04

**Soundness:** 3
**Presentation:** 4
**Contribution:** 2
**Rating:** 3
**Confidence:** 3

**Summary:**

The paper considers the task Fill-in-the-Middle (FIM) in code generation. Instead of completing the code, which is a common task in code generation, the authors consider the task of completing missing code given the left and right contexts.

They consider adding an auxiliary objective, predicting the number of missing lines, to the training of a language model. The model is trained to optimize next-token prediction as well as the number of missing lines. Empirically, this method improves performance on file-level and repository-level benchmarks.

**Strengths:**

FIM is an important problem that is relatively underexplored in the literature. The method proposed in this paper is simple and straightforward. It shows significant improvements on diverse benchmarks.

The paper is well-written and easy to follow.

**Weaknesses:**

**Contributions.**

Generalization: The proposed method appears to target at the FIM task. It limits its generalization to other code generation tasks.

Cost: The method requires fine-tuning a model specifically for FIM, which could be costly. Whenever a code model or a generalized model is released, it needs to be finetuned and maintained solely for this task.

**Evaluation metrics.** Table 4 uses Exact Match (EM) and Edit Similarity (ES) as evaluation metrics, which are not standard in code generation. This choice seems to be consistent with prior work. Is it possible to evaluate using pass@1 / pass@k? Or is it justifiable to measure EM and ES for codes?

**Questions:**

My questions are in the weaknesses section above. The authors are welcome to correct any possible misunderstandings.

---

> ### Author Response · Authors · 2024-11-21
>
> Thank you for the valuable comments! Our responses are as follows:
>
> > Summary: … They consider adding an auxiliary objective, predicting the number of missing lines, to the training of a language model …
>
> We would like to kindly clarify that HLP teaches models to predict the number of remaining future tokens (i.e., “missing tokens”) rather than missing lines.
>
> > W1: The proposed method appears to target at the FIM task. It limits its generalization to other code generation tasks.
>
> We appreciate your concern. We want to kindly highlight that FIM is a fundamental capability in modern code development. FIM is a standard pre-training task in state-of-the-art code LLMs [5,6,7,8,9,10], after being proposed by OpenAI [3] and Meta [4]. FIM has also powered widely-used features like inline code suggestions in GitHub Copilot [1,2], demonstrating its importance in the real world.
>
> > W2: The method requires fine-tuning a model specifically for FIM, which could be costly. Whenever a code model or a generalized model is released, it needs to be finetuned and maintained solely for this task.
>
> Thank you for your note. We would like to kindly clarify that HLP does not require additional fine-tuning. It is integrated into the standard code LM pre-training process where models will by default learn both left-to-right (L2R) and FIM capabilities simultaneously. In detail, we follow the standard code LM pre-training pipeline in our experiments by using 50% of the data for normal L2R pre-training and the other 50% for FIM pre-training (i.e., FIM ratio=0.5) [5,6,7,8,9,10]. HLP is directly incorporated into the FIM pre-training part, which has minimal impact on training efficiency as it only adds < 0.01% additional parameters.
>
> > W3: Table 4 uses Exact Match (EM) and Edit Similarity (ES) as evaluation metrics, which are not standard in code generation. This choice seems to be consistent with prior work. Is it possible to evaluate using pass@1 / pass@k? Or is it justifiable to measure EM and ES for codes?
>
> Thank you for your question. Our evaluation uses a complementary set of metrics appropriate for different scenarios following past literatures and standard practices:
>
> - For repository-level FIM tasks, we use EM/ES following standard practice in the literatures [11,12,13], because execution-based evaluation is often impractical due to complex dependencies and environment requirements in real-world repositories. We kindly refer the reviewers to the original CrossCodeEval [12] / CrossCodeLongEval [13] / RepoEval [11] papers for additional discussions.
> - For function-level FIM and code reasoning tasks where execution is feasible (i.e., SAFIM for FIM and CRUXEval for code reasoning), we report the standard execution-based pass@k metrics (Tables 2,3,5) following the literature [14,15].
> - For the code fixing task where execution is also feasible, we follow existing works [16,17] to report the number of generated patches that can successfully pass the test suite.
>
> These metrics should well capture the performance of the resulting models.
>
> Please feel free to let us know if any points remain unclear or if you would like us to elaborate further on any aspect of our responses.
>
> [1] Peng et al., 2023. The impact of ai on developer productivity: Evidence from github copilot.
>
> [2] Rosenkilde, J., 2023. How GitHub Copilot is getting better at understanding your code (https://github.blog/ai-and-ml/github-copilot/how-github-copilot-is-getting-better-at-understanding-your-code/)
>
> [3] Bavarian et al., Jun, H., Tezak, N., Schulman, J., McLeavey, C., Tworek, J. and Chen, M., 2022. Efficient training of language models to fill in the middle.
>
> [4] Fried et al., 2022. Incoder: A generative model for code infilling and synthesis.
>
> [5] Roziere et al., 2023. Code llama: Open foundation models for code.
>
> [6] Li et al., 2023. Starcoder: may the source be with you!.
>
> [7] Guo et al., 2024. DeepSeek-Coder: When the Large Language Model Meets Programming--The Rise of Code Intelligence.
>
> [8] Lozhkov et al., 2024. Starcoder 2 and the stack v2: The next generation.
>
> [9] Zhu et al., 2024. DeepSeek-Coder-V2: Breaking the Barrier of Closed-Source Models in Code Intelligence.
>
> [10] Hui et al., 2024. Qwen2. 5-coder technical report.
>
> [11] Zhang et al., 2023. Repocoder: Repository-level code completion through iterative retrieval and generation.
>
> [12] Ding et al., 2024. Crosscodeeval: A diverse and multilingual benchmark for cross-file code completion.
>
> [13] Wu et al., 2024. REPOFORMER: Selective retrieval for repository-level code completion.
>
> [14] Gong et al., 2024. Evaluation of LLMs on Syntax-Aware Code Fill-in-the-Middle Tasks.
>
> [15] Gu et al., 2024. Cruxeval: A benchmark for code reasoning, understanding and execution.
>
> [16] Xia et al., 2023. Automated program repair in the era of large pre-trained language models.
>
> [17] Jiang et al., 2023. Impact of code language models on automated program repair.

---

> ### Author Response · Authors · 2024-11-25
> **Looking forward to your feedback**
>
> Dear Reviewer s26Y,
>
> We greatly appreciate your time and expertise in reviewing our work. We have provided detailed discussions in our response based on your valuable comments.
>
> As the discussion period draws close, we greatly value your further input on our responses. We are committed to ensuring that our rebuttal aligns with your suggestions and incorporates your feedback to enhance the quality of our work.
>
> Once again, we extend our deepest gratitude for your insightful and valuable comments.
>
> Thank you and best regards.

---

> ### Comment · Reviewer_s26Y · 2024-11-26
>
> Thanks to the authors for the detailed answers to my questions and for correcting inaccuracies in my summary. I appreciate the authors' efforts in conducting experiments. However, I believe this is more of an ad-hoc design for a specific problem without a deeper explanation of why it works. I would keep my score the same.

---

> ### Author Response · Authors · 2024-11-27
> **Thank you for the response and let us further clarify the details**
>
> Thank you for your response! We would like to make further clarifications to address your concern on generalizability of our proposed method. In particular, we want to highlight that HLP is designed to be broadly *applicable for pre-training general-purpose code LLMs that can do both left-to-right generation and fill-in-the-middle*. In Appendix A.1, we discussed how to achieve *monotonic improvement* with a modified version of HLP, so that the resulting model shows superior performance on FIM tasks while retaining full capability on left-to-right (a.k.a. uni-directional) generation. Concretely, we apply HLP not only for FIM, but also for L2R training by asking the model to *predict the number of future tokens required to complete the current line*. The detailed evaluation results can be found in Table 8.
>
> Given the prevalence of FIM in code LMs and real-world code completion scenarios (which we discussed with references in our original response to W1), we believe the monotonic improvement on FIM without regression on left-to-right capability represents a major contribution of our work.
>
> To provide a deeper explanation of why HLP works, we demonstrate the connection between “planning” and horizon awareness by designing a specific decoding algorithm that can manipulate the verbosity of the solution during test time using HLP predictions.  Please feel free to check the examples in our latest response to Reviewer `gJnh` for full details.
>
> Once again we greatly appreciate your feedback. Kindly please let us know if there’s anything that we can clarify further, and we are more than happy to continue the discussion.

---

> ### Author Response · Authors · 2024-12-01
> **Looking forward to further discussion!**
>
> Dear Reviewer s26Y,
>
> Thank you for your response! As the discussion period deadline approaches, we would greatly appreciate your feedback on our further clarifications for your remaining concerns.
>
> Your feedback would greatly contribute to our work and the ICLR community!
>
> Thank you and best regards.

---

### Official Review · Reviewer_gJnh · 2024-11-04

**Soundness:** 2
**Presentation:** 3
**Contribution:** 2
**Rating:** 6
**Confidence:** 4

**Summary:**

The paper highlights the current limitations of Fill-in-Middle (FIM) evaluations of popular coding benchmarks due to the post-process that takes places, which often boost evaluation performance artificially. The main contribution of the paper is introducing a new auxiliary loss that predicts the "horizon" length (i.e. the portion of tokens left to predict) on top of the typical Prefix-Suffix-Middle next token prediction loss to perform FIM tasks. With this training objective, LLMs perform better at FIM tasks, notably, the repository-level cross-fill code tasks.

**Strengths:**

- The paper soundly points out a glaring problem with current evaluation of FIM tasks that convincingly concludes that the post-processing only artificially boosts model scores while not providing additional insights into performance towards practical settings
- The paper provides a simple but novel idea that shows model improvements in the more rigorous evaluation (without any post-processing) of FIM tasks that would be of interest for the research community to apply to other domains
- The paper explores a breadth of different evaluation tasks for assessing their methods, including direct FIM tasks, repository-level cross-file FIM tasks, code fixing tasks, and reasoning tasks, demonstrating the effectiveness of their method
- That paper is well written and easy to understand/follow

**Weaknesses:**

- Lack of rigorous confidence interval analysis - all the of experimental results lack statistical significance numbers, making it hard to judge if the performance improvements are due to noise or if they are statistically significant.
- Lack of theoretical/empirical evidence for why HLP works - it is not clear to me why this method works (assuming the experimental results are statistically significant). I believe the authors should add a section explaining (at least intuitively) why this method should work.
- Lack of additional baselines + ablations - adding some other strong baseline results would further validate this method. For instance, authors mention multi-token prediction in their related works. This methods performance should be reported as a strong baseline. For ablations, one idea could be explore the affect of increasing the complexity of the $hlp$_$head$ (e.g. using a MLP of increasing layers).

**Questions:**

- Could you report the statistical significance for each of the evaluations?
- I found Section 5 particularly interesting and believe a natural question to ask is if "Horizon Awareness" under NTP increases with model parameters?
- Why is $HLP_{L2R} + HLP_{FIM}$ not presented as the standard approach?
- Is there a reason that during training the HLP objective is applied for each token instead of just the first token? My concern is that remaining tokens after the first token do not really provide any "additional" signal for the horizon length.

---

> ### Author Response · Authors · 2024-11-21
>
> Thank you for your comprehensive review and thoughtful questions! We address your concerns as follows.
>
> > W1: Rigorous confidence interval analysis / Q1: Statistical significance for each of the evaluations
>
> Thanks for the suggestions! We have conducted a comprehensive statistical analysis to validate our findings. Specifically, we conduct paired t-tests between the baseline (NTP only) and our approach (NTP+HLP) across all benchmarks and report p-values to assess statistical significance.
>
> As shown in the table, our analysis reveals strong statistical significance for the majority of our evaluations, demonstrating the significant improvements brought by HLP. For Defects4J, we see a slightly higher p-value but we note that Defects4J is a small dataset with only 313 examples and models can only solve 30-60 out of these, and the limited number of samples naturally leads to higher variance in statistical estimates.
>
> | Model | SAFIM | CrossCodeEval / CrossCodeLongEval (EM / ES) | RepoEval (EM / ES) | Defects4J | CRUXEval |
> | -------- | -------- | -------- | -------- | -------- | -------- |
> | DS-1.3B | 3.530e-18 | 5.170e-7 / 3.381e-19 | 2.299e-6 / 1.692e-7 | 0.067 | 1.233e-9 |
> | DS-6.7B | 0.033 | 0.039 / 0.037 | 2.709e-6 / 6.613e-7 | 0.391 | 0.115 |
> | SC2-3B | 7.589e-21 | 4.521e-27 / 2.036e-64 | 3.845e-15 / 5.739e-22 | 0.297 | 0.001 |
> | SC2-7B | 5.756e-13 | 3.023e-14 / 3.280e-41 | 4.885e-17 / 1.539e-21 | 0.090 | 0.004 |
>
> > W2: Theoretical/empirical evidence for why HLP works
>
> We appreciate this important question. The effectiveness of HLP can be explained as follows:
>
> Intuitively, next-token prediction (NTP) is myopic in the sense that it does not take into account future constraints when predicting the immediate next token, as it only optimizes for local token-by-token decisions. While this is less of a problem for left-to-right code generation, in FIM there is always such a constraint that the middle part must connect smoothly to the right context. Standard NTP only tackles this constraint at the very last <eoi> token. However, at test time, the <eoi> token can sometimes be completely missing due to accumulated errors in preceding tokens which are not trained with any end-of-insertion information. In contrast, with HLP, every token gets a sense of how far away it is from connecting to the right context (i.e., global sequence-level planning), leading to a more reliable closing of the infilling generation. In other words, while NTP only provides a sparse signal from <eoi> token prediction at the end of generation, HLP enables a consistent optimization by providing an auxiliary learning signal at every generation step.
>
> Empirically, this theoretical advantage translates to practical benefits. We empirically verified our method across a wide range of coding tasks and demonstrated its statistically significant effectiveness (see above).
>
> We hope this discussion is clearer and we will expand this discussion in the paper to make these connections explicit. Thanks again for the suggestion.
>
> > W3: Additional baselines + ablations
>
> Thank you for your suggestions! We have conducted an additional experiment to compare HLP with multi-token prediction (n=4) on DeepSeekCoder-1.3B (same for other additional experiments). We report the performance on SAFIM. As shown in the table, while adding HLP largely improves the model’s performance on SAFIM, multi-token prediction fails to do so, indicating that multi-token prediction is not effective enough for FIM. As discussed in Section 6, the reason behind this is that multi-token prediction only enhances models’ planning capability for a short and limited horizon, which is usually insufficient for FIM as the connection from middle to suffix happens over a long horizon.
>
> | | Algorithmic | Algorithmic_v2 | Control | API | Average |
> | -------- | -------- | -------- | -------- | -------- | -------- |
> | DS-1.3B | 39.8 | 42.4 | 52.4 | 56.1 | 47.7 |
> |  + HLP | **41.3**| **46.1** | **53.4** | **59.0**  | **50.0** |
> |  + multi-token prediction | 38.1 | 41.3 | 51.7 | 55.8 | 46.7 |
>
> As suggested by the reviewer, we have also conducted another experiment to study the effect of the complexity of the $hlp$\_$head$ by replacing the original linear layer (i.e., “HLP (linear)”) with a two-layer MLP with ReLU (i.e., “HLP (mlp)”). As shown in the table, increasing the complexity of $hlp$\_$head$ does not bring significant improvements. We have also conducted a paired t-test between “HLP (linear)” and “HLP (mlp)” and did not see any clear directional statistical significance between them. Such results indicate that the complexity of $hlp$\_$head$ does not have a major impact on performance.
>
> | | Algorithmic | Algorithmic_v2 | Control | API | Average |
> | -------- | -------- | -------- | -------- | -------- | -------- |
> | DS-1.3B | 39.8 | 42.4 | 52.4 | 56.1 | 47.7 |
> |  + HLP (linear) | 41.3 | **46.1** | 53.4 | **59.0**  | **50.0** |
> |  + HLP (mlp) | **41.6** | 45.9 | **54.0** | 57.4 | 49.7 |

---

> > ### Author Response · Authors · 2024-11-21
> >
> > > Q2: Does "Horizon Awareness" under NTP increase with model parameters?
> >
> > Great question and thank you for your interest! We have conducted additional probing experiments on all the models, including DeepSeekCoder 1.3B/6.7B and StarCoder2 3B/7B. As shown in the table, “Horizon Awareness” under NTP indeed increases with more model parameters. However, for all the models, “Horizon Awareness” with NTP only is still very weak, showing that training with NTP only cannot bring “Horizon Awareness” to language models effectively, even with a larger model.
> >
> > | | Training | Test |
> > | -------- | -------- | -------- |
> > | DS-1.3B | 0.455 | 0.440 |
> > |  + HLP | **0.919**| **0.915** |
> > | DS-6.7B | 0.525 | 0.519 |
> > |  + HLP | **0.919**| **0.913** |
> > | SC2-3B | 0.364 | 0.356 |
> > |  + HLP | **0.927**| **0.932** |
> > | SC2-7B | 0.418 | 0.410 |
> > |  + HLP | **0.929**| **0.932** |
> >
> > > Q3: Why is $HLP_{L2R}+HLP_{FIM}$ not presented as the standard approach?
> >
> > Great question! Please kindly note that our work focuses on the problem of FIM code completion. Consequently, we present $HLP_{FIM}$ as the standard approach in our main paper.
> >
> > Applying the same idea of HLP to L2R is definitely interesting to explore. However, the “horizon length” in L2R is not well-defined. As discussed in the appendix, the major difference between FIM and L2R is that, while the length of the middle in FIM is strictly bounded by the beginning of the suffix, there is no such clear boundary in L2R. One may think of “end of file (EOF)”, but we argue that EOF as a boundary can be ambiguous as it is often possible to add additional contents (e.g., another class or a new helper function) to the end of document fluently without any restrictions. Therefore, we exercised our first attempt with line-level boundaries for L2R (as shown in the appendix) and left further explorations as future work.
> >
> > > Q4: Is there a reason that during training the HLP objective is applied for each token instead of just the first token?
> >
> > Great question! While we fully agree that knowing the HLP loss on the first token is sufficient to infer the horizon length, having HLP loss on every token provides denser and consistent supervision signals which makes learning easier (see theoretical discussion above). It also helps regularize the hidden representation of every subsequent token. To empirically show this, we have conducted an additional experiment by only applying HLP to the first token only (i.e., “HLP (first)”) and compared its performance with our original HLP design (i.e., “HLP (all)”). As shown in the table, while applying HLP only to the first token performs better than NTP only, applying HLP loss for each token can achieve better performance than applying it to just the first token.
> >
> > |  | Algorithmic | Algorithmic_v2 | Control | API | Average |
> > | -------- | -------- | -------- | -------- | -------- | -------- |
> > | DS-1.3B | 39.8 | 42.4 | 52.4 | 56.1 | 47.7 |
> > |  + HLP (all) | **41.3**| **46.1** | **53.4** | **59.0**  | **50.0** |
> > |  + HLP (first) | 40.0 | 44.4 | 52.3 | 57.4 | 48.5 |

---

> ### Author Response · Authors · 2024-11-25
> **Looking forward to your feedback**
>
> Dear Reviewer gJnh,
>
> We greatly appreciate your time and expertise in reviewing our work. We have provided additional experiments, theoretical explanations, and detailed discussions in our response based on your valuable comments.
>
> As the discussion period draws close, we greatly value your further input on our responses. We are committed to ensuring that our rebuttal aligns with your suggestions and incorporates your feedback to enhance the quality of our work.
>
> Once again, we extend our deepest gratitude for your insightful and valuable comments.
>
> Thank you and best regards.

---

> ### Comment · Reviewer_gJnh · 2024-11-26
>
> I thank the authors for their detailed responses. All of my questions have been addressed and it has strengthened my understanding of this paper. I particularly like the simplicity yet effectiveness of this method and particularly convinced by the additional horizon-awareness of larger model parameters experiments. I am still not fully convinced by the intuitive/theorectical explanation of why this method works. In particular, I don't understand the connection between "planning" and horizon awareness. It would be interesting to see if, during test time, you can manipulate the verbosity of the solution by adjusting the "length label" for the FIM chunk. If you can, this would convince me that the model is sufficiently adjusting its "plan". Overall, I appreciate the authors detailed response, and I have decided to raise my score and lean towards accepting this paper.

---

> > ### Author Response · Authors · 2024-12-01
> > **Looking forward to further discussion!**
> >
> > Dear Reviewer gJnh,
> >
> > Thank you for your response! As the discussion period deadline approaches, we would greatly appreciate your feedback on our further discussion about the connection between HLP and planning.
> >
> > Your feedback would greatly contribute to our work and the ICLR community!
> >
> > Thank you and best regards.

---

> ### Author Response · Authors · 2024-11-27
> **Thank you for one more great suggestion!**
>
> Thank you for your response and support!
>
> You raised a great point on verbosity control at test time. To demonstrate this, we design a specific decoding algorithm that can manipulate the verbosity as follows:
> 1. For a given prompt, sample only the first 3 tokens with a high temperature (i.e., $t=1$) for multiple times. This ensures we have diverse generations from the same prompt.
> 2. For each sampled 3-token prefix, predict the length of the subsequent generation using $hlp$\_$head$.
> 3. Decide which 3-token prefix to use based on the length prediction, and generate the whole answer with greedy decoding given the original prompt and the chosen 3-token prefix.
>
> Given this decoding algorithm, we are able to manipulate the verbosity of the solution by using HLP predictions to choose the opening tokens that can lead to longer or shorter solutions during test time, without actually generating the complete solutions. This decoding algorithm is also efficient, as sampling only happens at the first three tokens.
>
> We show a qualitative example using a problem randomly picked from SAFIM, where the target is to fill in the code to replace `{{completion}}`:
> ```
> # SAFIM block_completion_004203
> def solve():
>     n = int(input())
>     a = [int(i) for i in input().split()]
>     dl, dr = 0, 0
>     for i in range(1, n):
>         if a[i]-dr >= a[0]-dl:
>             dr += (a[i]-dr)-(a[0]-dl)
>         else:
>             {{completion}}
>     return dl+dr+abs(a[0]-dl)
>
> for _ in range(int(input())):
>     print(solve())
> ```
>
> For each sampled 3-token prefix, we provide both the HLP prediction and the subsequent completion solution for a clear demonstration. As shown in the following example generations, HLP predictions align well with the actual length of the complete solutions, indicating that the verbosity of the solution can be manipulated during test time with our proposed decoding algorithm.
>
> 1. First three token: `dl +=  abs`
>     * Predicted length using $hlp$\_$head$: 16.00
>     * Actual length: 10
>     * Complete solution:
> ```
>             dl += abs(a[i]-dr)
> ```
> 2. First three token: `dl += -`
>     * Predicted length using $hlp$\_$head$: 22.26
>     * Actual length: 18
>     * Complete solution:
> ```
>             dl += -(a[i]-dr)+(a[0]-dl)
> ```
> 3. First three token: `d = a`
>     * Predicted length using $hlp$\_$head$: 37.24
>     * Actual length: 34
>     * Complete solution:
> ```
>             d = a[i]-dr
>             dl += d
>             dr += d
>     dl += abs(a[0]-dl)
> ```
>
> Thank you once again for suggesting this great idea to further our understanding of the connection between HLP and planning, and we will add this discussion into the paper. Please do not hesitate to let us know if you have any other questions or suggestions.

---

### Author Response · Authors · 2024-11-21
**Summary of Responses**

We thank all the reviewers for their insightful comments and suggestions for improving the paper! We have conducted additional ablations and analyses following reviewers’ suggestions, and we address the main weaknesses and questions in the response to individual reviewers below. We will add these ablations, analysis, and discussions to a later revision of the paper.

Please let us know if you have additional questions or need more clarification for our responses. We are happy to discuss further with reviewers during the reviewer-author discussion period.

---

### Author Response · Authors · 2024-11-24
**Submission Revision**

We thank the reviewers for the inspiring comments and we have updated our submission by making the following notable changes (highlighted in blue):

- **(Experiment) Adding statistical significance for each of the evaluations (Section 4 / Table 3,4,5)**: our analysis reveals strong statistical significance for the majority of our evaluations. (gJnh)

- **(Experiment) Adding multi-token prediction as a baseline (Section 5.1 / Table 6)**: multi-token prediction performs much worse than HLP due to limited planning horizon. (gJnh)

- **(Experiment) Conducting additional probing experiments on all the models (Section 5.2 / Table 7 / Figure 4)**: while “Horizon Awareness” under NTP increases with more model parameters, they are still very weak, showing that training with NTP only cannot bring “Horizon Awareness” to language models effectively, even with a larger model. (gJnh)

- **(Experiment) Studying the effect of the complexity of `hlp_head` (Appendix A.2 / Table 9)**: we find that the complexity of `hlp_head` does not have a major impact on performance. (gJnh)

- **(Experiment) Comparing against applying HLP to the first token only (Appendix A.2 / Table 10)**: we find that applying HLP loss for each token can achieve better performance than applying it to just the first token, as having HLP loss on every token provides denser and consistent supervision signals which makes learning easier. (gJnh)

- **(Experiment) Comparing against using `M-t` as the target in HLP (Appendix A.2 / Table 11)**: we find that setting the target as `M-t` fails to improve FIM performance of baselines, likely due to the large-scale HLP loss after using `M-t` as the target interferes with NTP pre-training. (ogj5)

- **(Discussion) Providing a preliminary theoretical explanation for HLP (Section 3).** (gJnh)

- **(Writing) Fixing the typo (Section 2).** (ogj5)

We are looking forward to discussing further with reviewers!

---

### Author Response · Authors · 2024-12-04
**Summary of Author-Reviewer Discussion**

We thank all the reviewers for their thoughtful feedback and constructive suggestions. Below, we summarize the main discussion points and how they were addressed through the rebuttal process:

**Statistical Significance**
- **Reviewer’s Concern**: Reviewer `gJnh` questioned whether the performance improvements of HLP are statistically significant.
- **Our Response**: We performed additional statistical significance analysis and revealed strong significance (p<0.05 through paired t-test) across most evaluations.

**Design Choices & Ablations**
   - **Reviewer’s Concern**: Reviewer `gJnh` questioned (1) whether the complexity of `hlp_head` affects the effectiveness of HLP, (2) whether applying HLP to just the first token can achieve similar performance as applying HLP for each token, and (3)  whether HLP can outperform multi-token prediction. Additionally, Reviewer `ogj5` questioned the necessity of using `M-t/M` rather than `M-t` as the training target.
   - **Our Response**: We performed ablation studies for all these asks. We show that:
        - the complexity of `hlp_head` has a minimum impact on HLP’s performance;
        - applying HLP for each token can achieve better performance than applying it to just the first token, as having HLP loss on every token provides denser and more consistent supervision signals which makes learning easier;
        - multi-token prediction performs much worse than HLP. The reason behind this is that multi-token prediction only enhances models’ planning capability for a short and limited horizon, which is usually insufficient for FIM as the connection from middle to suffix happens over a long horizon;
        - setting the target as `M-t` fails to improve the FIM performance of baselines, likely due to the large-scale HLP loss after using `M-t` as the target interferes with NTP pre-training.

        These results successfully justify the original design choices of HLP.

**Theoretical Understanding/Why HLP works**
  - **Reviewer’s Concern**: Reviewers `gJnh`, `s26Y`, and `ogj5` asked about the theoretical/intuitive explanation for why HLP works. Reviewer `gJnh` further questioned the connection between HLP and “planning”.
  - **Our Response**: We provided a preliminary theoretical explanation showing how HLP addresses myopic generation in standard next-token prediction. We demonstrated the concrete connection to planning through a novel decoding algorithm that enables verbosity control using HLP predictions (thank you Reviewer `gJnh` for the interesting idea!).

**Scaling of Horizon Awareness**
  - **Reviewer’s Concern**: Reviewer `gJnh` was curious about whether "Horizon Awareness" under NTP increases with model parameters.
  - **Our Response**: We performed additional experiments to reveal that, while "Horizon Awareness" increases with model size under NTP, it remains consistently weak without HLP, demonstrating the fundamental limitation of standard training.

**Generalizability**
   - **Reviewer’s Concern**: Reviewers `s26Y` and `ogj5` expressed concerns about limited generalization beyond FIM tasks and integration with uni-directional generation in code LLMs.
   - **Our Response**: We highlighted that:
        - FIM plays a fundamental role in modern code development and is a standard pre-training task in all code LLMs.
        - HLP is designed to be broadly applicable for pre-training general-purpose code LLMs that can do both left-to-right generation and fill-in-the-middle and demonstrates the monotonic improvement on FIM without regression on left-to-right capability, as supported by empirical evidence in Appendix A.1.

**Evaluation Metric**
- **Reviewer’s Concern**: Reviewer `s26Y` questioned whether Exact Match (EM) and Edit Similarity (ES) should be used as evaluation metrics for repository-level FIM tasks.
- **Our Response**: We note that it is the standard practice in the literature to use EM/ES for repository-level FIM tasks from multiple previous works in top conferences like NeurIPS/ICLR. We further highlighted that our evaluation uses a complementary set of metrics appropriate for different scenarios following past literature and standard practices, e.g., execution-based metrics for function-level tasks, and test suite pass rates for code fixing, which well captures the performance of the resulting models.

Finally, we sincerely thank all reviewers again for their dedication and thoughtful evaluations, and we are also deeply appreciative of the Area Chair’s guidance and support.

Thanks,

Authors of HLP

---

### Meta-Review · Area_Chair_MsLm · 2024-12-21

**Metareview:**

This paper introduces horizon-length prediction, a novel training objective designed to improve fill-in-the-middle tasks by enabling models to predict the number of remaining middle tokens, thus facilitating effective planning based on distant right context. Despite of the contributions, the paper still needs to be improved on its lack of statistical significance analysis, insufficient justification for the method's effectiveness, limited generalization, high fine-tuning costs, reliance on simple techniques, questionable target setting, and inadequate analysis of code generation length prediction.

**Additional Comments On Reviewer Discussion:**

Generally the discussions are thorough.

---

### Decision · Program_Chairs · 2025-01-22

Reject